# MApLe: Multi-instance Alignment of Diagnostic Reports and Large Medical Images

**Felicia Bader**[1,2]      FELICIA.BADER@MEDUNIWIEN.AC.AT

**Philipp Seeböck**[1,2,3]      PHILIPP.SEEBOECK@MEDUNIWIEN.AC.AT

**Anastasia Bartashova**[4]      ANASTASIA.BARTASHOVA@MEDUNIWIEN.AC.AT

**Ulrike Attenberger**[2,4]      ULRIKE.ATTENBERGER@MEDUNIWIEN.AC.AT

**Georg Langs**[1,2]      GEORG.LANGS@MEDUNIWIEN.AC.AT

[1] *Computational Imaging Research Lab, Department of Biomedical Imaging and Image-guided Therapy, Medical University of Vienna, Austria*

[2] *Comprehensive Center for Artificial Intelligence in Medicine, Medical University of Vienna, Austria*

[3] *Medical Anomaly Detection (MANO) Group, Computational Imaging Research (CIR), Department of Biomedical Imaging and Image-guided Therapy, Medical University of Vienna, Austria*

[4] *Department of Biomedical Imaging and Image-Guided Therapy, Medical University of Vienna, Austria*

**Editors:** Accepted for publication at MIDL 2026

## Abstract

In diagnostic reports, experts encode complex imaging data into clinically actionable information. They describe subtle pathological findings that are meaningful in their anatomical context. Reports follow relatively consistent structures, expressing diagnostic information with few words that are often associated with tiny but consequential image observations. Standard vision language models struggle to identify the associations between these informative text components and small locations in the images. Here, we propose "MApLe", a multi-task, multi-instance vision language alignment approach that overcomes these limitations. It disentangles the concepts of anatomical region and diagnostic finding, and links local image information to sentences in a patch-wise approach. Our method consists of a text embedding trained to capture anatomical and diagnostic concepts in sentences, a patch-wise image encoder conditioned on anatomical structures, and a multi-instance alignment of these representations. We demonstrate that MApLe can successfully align different image regions and multiple diagnostic findings in free-text reports. We show that our model improves the alignment performance compared to state-of-the-art baseline models when evaluated on several downstream tasks. The code is available at https://github.com/cirmuw/MApLe.

**Keywords:** Vision-Language Alignment, Multi-instance Learning

## 1. Introduction

Unsupervised alignment of image and text representation methods have demonstrated strong performance in image classification and other downstream tasks (Radford et al., 2021b; Desai and Johnson, 2021; Li et al., 2020; Wang et al., 2021; Chen et al., 2020). They aim to align image representations with representations from their corresponding description or caption as an unsupervised pre-training method to improve the extraction of

meaningful image features. Traditional contrastive learning aligns images with clearly distinct captions (Radford et al., 2021a; Ma et al., 2025; Yao et al., 2021a; Gao et al., 2022; Li et al., 2021). In contrast, medical reports from the same image modality and clinical task often show a certain semantic consistency, where the same anatomical regions or findings are described with similar phrasings across different reports that vary only in subtle yet diagnostically significant details. These fine-grained text differences are crucial when identifying and linking diagnostically relevant image content. When aligning reports, sentences indicating the same diagnostic meaning should be recognized as similar rather than distinct despite originating from different reports and potentially differing in the used vocabulary. Additionally, medical reports consist of several sentences rather than one caption, therefore it is necessary to align image regions to a variety of text passages, each describing a different anatomical or diagnostic concept in the image. Since these concepts can have different sizes in the image, the image encoder will have to be capable of recognizing fine-grained structures as well as large ones.

To tackle these challenges, we propose a novel method for aligning medical image representations with their associated radiology reports. Our approach consists of three components: *1) Report Embedding:* During pre-training, we use keyword search to assign each sentence to a finding (e.g., calcification, aorta ...), and fine-tune the textual embedding model BERT (Devlin et al., 2019) to amplify fine-grained differences between sentences for each finding, thus deforming the feature space to better reflect semantically subtle but diagnostically relevant variations. *2) Image Encoding:* We automatically segment anatomical regions and embed them separately. Each region is split into patches, with the size of patches related to the scale of potential pathologies. *3) Multi-instance Alignment:* We align images and report features using a triplet loss, matching sentences in a report with the anatomical region in the image that represents its finding. The triplet loss minimizes the distance between this region and the corresponding sentence, as well as sentences from other reports that are semantically similar. On the other hand, it maximizes the distance between the region and sentences describing a different diagnostic meaning.

This approach learns to encode images to a similarity structure mirroring the diagnostically relevant localized structures in the reports. After the alignment, our model can be used for zero-shot multi-task classification of the described findings in the report. We evaluate our method using a 3D cardiac CT dataset across multiple downstream classification tasks.

## 2. Related Work

**Vision-Language-Alignment:** The alignment of image and text follows the aim of improving image feature extraction by bringing image and text features into the same latent space and maximizing the similarity between corresponding images and text. This type of pre-training has been tested for image classification (Radford et al., 2021b; Desai and Johnson, 2021; Jia et al., 2021; Yao et al., 2021b; Yu et al., 2022), image captioning (Li et al., 2020; Wang et al., 2021; Yu et al., 2022), visual question answering (VQA) (Chen et al., 2020; Li et al., 2020; Wang et al., 2021) or image-text-retrieval (Chen et al., 2020; Li et al., 2020; Jia et al., 2021; Yao et al., 2021b; Yu et al., 2022). One of the most popular approaches is CLIP (Radford et al., 2021a), where the model is trained to correctly predict pairings of images and text. This model is designed for aligning real-world RGB images

with short, distinct captions, making it great for feature learning and image captioning in a real-world setting. ConVIRT (Zhang et al., 2022) applies the alignment strategy to medical 2D images and their respective radiology reports and achieves decent classification results when pairing 2D chest radiographs with their report texts. BioVIL (Boecking et al., 2022) introduced an improved vocabulary and pre-training for their text encoder in order to enhance the contrastive learning process. The extension BioVIL-T (Bannur et al., 2023) further improves performance by also accounting for prior images which can be referred to in the clinical report. LoVT (Müller et al., 2022) combines global image-report alignment with local sentence-region alignment. The model GLoRIA (Huang et al., 2021) tries to enhance local features in the image by splitting the report into words and aligning image sub-regions with these words. Their classification approach also works in a zero-shot setting. However, this zero-shot classification does not work in a multi-task setting where images are allowed to have more than one positive label for the classification tasks. Another method has been established that also utilized multiple instance learning in order to align image and text features by combining local and regional features (Wang et al., 2023). However, their approach splits the whole image into non-overlapping patches and matches all patches with all sentences in the report. In contrast, MApLe aligns patches and sentences based on anatomical or diagnostic findings, making it easier to align fine-grained structural differences in large scale 3D images with their respective sentence.

All of these methods were designed for 2D images like chest X-rays with their respective reports. When using 3D images like CT volumes, the images get more complex with a large volume and only relatively small structural differences. A computational efficient image encoder has to be constructed that is still capable of detecting these structural differences in large volumes. Additionally, the reports also get more complex when describing not only a single X-ray slide but a 3D volume with multiple anatomical structures and diagnostic findings. The above mentioned methods lack the capability of aligning a large 3D volume with several sentences where each sentence describes a different diagnostic aspect about the image. Moreover, the text encoder also has to be capable of differentiating between different clinical findings. Foundation text encoders like used in the methods above are pre-trained on a large corpus of medical texts. Radiology reports describing a specific organ-system therefore only get a small area in the feature space, making sentences from these reports almost indistinguishable.

**Multi-Instance Learning:** Multi-instance learning describes the phenomenon that for each sample you have a bag of data points that are not individually labeled. It is only known for the overall sample which class it belongs to. During training, bag-level labels are utilized, i.e. the model looks at the instances in the bag and learns to predict the overall bag label. In case the overall bag is positive, some instance in the bag has to give back a positive feedback. If the overall bag is negative, all instances in the bag should be negative. This approach has been extensively utilized when training a classifier where each sample consists of several instances in order to reduce the labeling effort (Ilse et al., 2018; Qu et al., 2022; Tang et al., 2023), or even for image captioning or classification based on images and corresponding captions (Zhou et al., 2024).

We propose MApLe to overcome the drawbacks of the above mentioned models by (1) fine-tuning the text encoder to enhance fine-grained differences in sentences, (2) introducing a patch based image encoder capable of working with high resolution 3D volumes and (3) a

multi-task, multi-instance alignment process that can align one image to several sentences, each describing a different diagnostic aspect about the image. Our approach is capable of a zero-shot classification concerning the defined diagnostic findings.

## 3. Methodology

We consider pairs of images and reports $\langle \mathbf{I}, R \rangle$ where $\mathbf{I} \in \mathbb{R}^{k \times l \times h}$ is a three-dimensional image volume, from which we can sample local patches $\mathbf{p}$. $R = \langle s_1, s_2, ..., s_n \rangle$ is a report consisting of a sequence of sentences. A sentence describes a diagnostic finding $d$ located in an anatomical region. The finding has a class $c$ (e.g., positive or negative, i.e., a coronary artery does or does not contain calcification). We learn an image patch embedding $f_a$ : $\mathbf{p} \mapsto \vec{x}$, dependent on the anatomical region $a$ of the patch, and a sentence embedding $g_d : s \mapsto \vec{y}$, dependent on the finding $d$ the sentence describes. For each diagnostic finding, we then learn projections of the intermediate latent image representations $(\vec{x})$ into the text embedding space: $a_I^d : \vec{x} \mapsto \vec{x}_a^d$, so that report descriptions of findings are aligned with the instances of findings in the image. This is a multi-instance learning problem, as findings are typically localized and cover only a small part of an anatomical image region. After training, we can embed each image patch of a new image by all $f_a$. Figure 1 provides an overview of the approach.

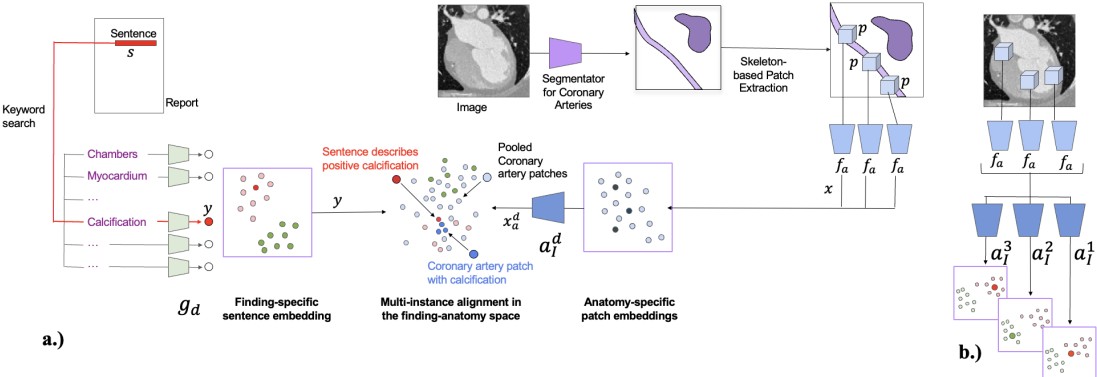

Figure 1: Multi-instance alignment of diagnostic reports and image volumes. (a) During training the report is decomposed into sentences, each describing a finding, and its corresponding class (e.g., present or not). Anatomical regions in images are segmented, and patches are extracted. A finding specific multi-instance alignment of sentence- and image representations maps textual finding classes and associated image patches into a joint embedding space. (b) During test time, all patches of an anatomical region are first encoded and than pooled together using the corresponding finding network. The resulting profile describes the diagnostically relevant findings.

### 3.1. Sentence Embedding

We learn sentence embeddings $g_d : s \mapsto \vec{y}$ that resolve clinically relevant observations in reports. We use the findings $d$ (e.g., calcification, aorta, ...), and their state $c$ (present, dilated) to guide the embedding. During training, we first label sentences based on the identified findings and their states. For each finding, we then train an embedding model by composing a model $h_d \circ g_d$, where we initialized $g_d$ with a BERT (Devlin et al., 2019) sentence embedding, and $h_d$ is a classification head that consists of a single linear layer. We train the composed model to predict the labeled finding state $c$, fine-tuning both $g_d$, and $h_d$. In practice this labeling of finding and state can be performed by key-word matching. The resulting text encoder $g_d$ represents the variability of how a specific finding is described. For new sentences, the finding described in it is identified with the same key-word matching. Afterwards, the sentence is given into the corresponding text encoder to generate an embedding as well as to classify the state of the finding. A sentence is therefore mapped followed:

$$
\begin{aligned}
g_d &: s \mapsto \vec{y} \\
h_d &: \vec{y} \mapsto c.
\end{aligned}
\tag{1}
$$

We get the embedding $\vec{y}$ which is used for the alignment training. The number of encoders $g_d$ and classifiers $h_d$ corresponds to the number of findings present in the reports.

### 3.2. Image Embedding

Similarly to the sentence embedding, we learn an image patch embedding $f_a : \mathbf{p} \mapsto \vec{x}$ depended on the anatomical region of the patch $a \in A = \{arteries, myocardium, ...\}$. The identification of anatomical regions is based on a segmentation of the entire image volume. Each encoder receives patches for a region and generates an encoding for each patch.

### 3.3. Multi-Task Multi-Instance Alignment

We align the patch embeddings $\vec{x}$, and sentence embeddings $\vec{y}$ by using small attention networks $a_I^d$ conditioned on the finding reported in the sentence, yielding $\vec{x}_a^d$. These attention networks pool all patches from the anatomical region together to create one embedding. Note that this means that a set of patches can generate multiple aligned embeddings, based on which sentence they are aligned to. During test-time, we can view this as *probing* of the image content, to test its relationship to each possible finding that can occur in its anatomical region. This is a multi-instance learning problem, since we only know that a bag of patches is positive or negative but not in which patches a possible finding occurs. Multi-task alignment means that we want to align one image to a set of sentences where each sentence describes a different diagnostic finding with a specific state. The number of tasks is defined by the number of diagnostic findings in the report. For the multi-task alignment, a set of attention networks for patches from each anatomical region $Q_a = [a_I^1, a_I^2, ..., a_I^d,]$ is defined, where the number of networks corresponds to the number of diagnostic findings from the reports that can occur in the anatomical region. This enables the simultaneous multi-task alignment and aims to project the patch embeddings into the same feature space as the

sentence embeddings. The patch embedding for the alignment looks as followed:

$$
\begin{aligned}
f_a &: p_a \mapsto \vec{x} \\
a_I^1 &: \{\vec{x}\} \mapsto \vec{x}_a^1 \\
&\quad ... \\
a_I^d &: \{\vec{x}\} \mapsto \vec{x}_a^d,
\end{aligned}
\tag{2}
$$

when the anatomical region $a$ corresponds to $d$ diagnostic findings defined in the report. We now want to maximize the similarity between an aggregated embedding and its corresponding sentence in the report and all other sentences from other reports that describe the same diagnostic finding and state. This is achieved by utilizing the triplet loss (Schroff et al., 2015). Suppose we have a m-dimensional sentence representation $\vec{y} \in \mathbb{R}^m$, the corresponding finding of this sentence $d$ and also the state $c$ of this finding. The sentence comes from a report that has a corresponding image with embeddings depending on the anatomical region and diagnostic finding. Now, we take the embedding $\vec{x}_a^d$ where the previous attention mechanism $a_I^d$ corresponds to the diagnostic finding $d$ of the sentence embedding $\vec{y}$. This embedding is our anchor in the triplet loss. For the positive and negative point of the triplet loss, we utilize a set of sentence embeddings $\{\vec{y}\}_{pos}$ that all have the same diagnostic finding $d$ and state $c$ as the original sentence and another one $\{\vec{y}\}_{neg}$, where all embeddings have the same diagnostic finding $d$ but a different state $c$ than the original sentence. We now select a positive embedding that is dissimilar to the anchor vector, i.e. an embedding that has a relatively low cosine similarity with the image embedding:

$$
\vec{y}_p \sim P(\{\vec{y}\}_{pos}) \mid \vec{x}_a^d) \propto \exp(\text{-sim}(\{\vec{y}\}_{pos}, \vec{x}_a^d)),
\tag{3}
$$

with $\vec{y}_p \in \{\vec{y}\}_{positive}$. We select dissimilar embeddings with a higher probability since the anchor vector should be close to all embeddings in the positive set. Additionally, a similar negative embedding

$$
\vec{y}_n \sim P(\{\vec{y}\}_{neg}) \mid \vec{x}_a^d) \propto \exp(\text{sim}(\{\vec{y}\}_{neg}, \vec{x}_a^d)),
\tag{4}
$$

with $\vec{y}_n \in \{\mathbf{y}\}_{negative}$ is selected. Now, we select similar embeddings with a higher probability, to ensure that the anchor vector is pushed away from all embeddings in the negative set. The triplet loss is calculated as described by (Schroff et al., 2015)

$$
\mathcal{L} = \left[ \|\vec{x}_a^d - \vec{y}_p\|_2^2 - \|\vec{x}_a^d - \vec{y}_n\|_2^2 + \alpha \right]_+,
\tag{5}
$$

with a defined margin $\alpha$. This is done for every sentence in every report in the training dataset. During the alignment training, the text encoders $g_d$ and image encoders $f_a$ are not updated and only the weights in $a_I^d$ are optimized.

### 3.4. Multi-Task Zero-Shot Classification

For the image classification, we first encode the patches in the image as described above for each anatomical region $a$. We now suppose that we have a set of sentence embeddings $\{\vec{y}\}$ with corresponding state labels $\{c\}$ for each diagnostic finding $d$. The diagnostic findings

correspond to the classification tasks. For each finding $d$, the embedding $\vec{x}_a^d$ where the attention mechanism $a_I^d$ corresponds to the finding $d$ is selected. For this embedding the closest sentence embedding in $\{\vec{y}\}$ is calculated.

$$\vec{y}^* = \arg\max \ \text{sim}\left(\vec{x}_a^d, \{\vec{y}\}\right) \tag{6}$$

The state $c^*$ of this sentence embedding is assigned to the image embedding as the final classification label. This is a zero-shot classification, since no additional training for the classification of the diagnostic findings is necessary.

## 4. Experiments

### 4.1. Data

We trained our alignment model on an internal dataset from the Vienna General Hospital, consisting of 768 heart CT volumes and their corresponding radiology reports. The images come from a retrospective patient cohort from 2016 to 2023 that went through routine cardiac CT scans. It consists of two subgroups, one with a BMI over 30 and one with a BMI below 30. The CT images came from the SOMATOM Force scanner and we utilized the images recorded during the diastolic phase of the heart, i.e. the relaxation phase where the atria and ventricles are filled with blood. For the image acquisition, a contrast agent was utilized, dependent on patient characteristics. The initial thickness of the slices was at 0.6 and the resolution at $512 \times 512$ with different pixel spacings. For our approach, the images were pre-processed by resampling them to a uniform voxel size of $0.5 \times 0.5 \times 0.5$. Afterwards they were cropped around the heart with a target size of $512 \times 512 \times 512$ and normalized to values between 0 and 1. The reports were unstructured free-texts without any sections or fields. They contained technical information about the image acquisition, as well as the clinical information about the coronary arteries, aorta, pulmonary trunk and myocardium.

### 4.2. Experimental Details

**Image Encoder:** We utilize the segmentation of TotalSegmentator (Wasserthal et al., 2023) for the coronary arteries and the high resolution heart chambers segmentation. From the second setting we get the segmentation for the aorta, the pulmonary trunk and the myocardium. We define three anatomical regions: the coronary arteries, the myocardium and the aorta together with the pulmonary trunk, resulting in three patch encoders. The encoder for the coronary arteries accepts a patch size of $32 \times 32 \times 32$, the one for the myocardium and the aorta and pulmonary trunk utilizes $64 \times 64 \times 64$ patches. All encoders consist of 5 layers, where each layer consists of three convolution blocks with a residual connection. After these layers, a fully connected layer is introduced to map the embedding into a 768-dimensional feature space. The patches were sampled based on the segmentation. We used the sampling method, 'skeleton', where the segmentation is reduced to the middle-line. The points on this middle-line are used as centers for the extracted patches. The middle-line is walked over with a defined stride and the patches are extracted so that the middle-line is in the middle of the patch. The patch encoders are pre-trained on a reconstruction task. During this training, each patch was randomly augmented and with a

probability of 50% flipped along any axis or rotated. The number of epochs was set to 200 with a learning rate of 0.0001, a weight decay of 0.001 and the Adam optimizer and a batch size of 32. After this pre-training, the image encoder is used for the alignment training with frozen weights.

**Report Embedding:** The reports in our dataset came from the heart CT images. Since the reports are provided as unstructured free-texts without explicit sections, we apply sentence-level filtering based on task-specific keywords to retain clinically relevant descriptions while removing acquisition-related or administrative sentences. As diagnostic findings we defined calcification, stenosis, anomalies in the myocardium and a dilated aorta or pulmonary trunk (TP). These were also the keywords that were searched for when identifying the finding in the sentences. For each finding, two states are utilized, indicating if the phenomenon is present or not. The German BERT model was fine-tuned for the state classifications. The number of epochs was set to 200 with a learning rate of 0.0001, a weight decay of 0.001 and the Adam optimizer and a batch size of 32. The weights of the sentence encoder were frozen during the alignment training.

**Alignment Training:** During the alignment, both encoders are combined. For the image embedding, transformer networks were added in order to pool the embeddings from all patches. We implemented one transformer network for each diagnostic finding defined by the reports. The pooling for the calcification and stenosis were conducted for the patches from the coronary arteries. The patches from the myocardium and the aorta/pulmonary trunk were pooled once since only one diagnostic finding occurred in them. Each transformer network consisted of six layers, each with an embedding dimension of 768, 12 attention heads and a dropout with a probability of 0.1. For the alignment training, the number of epochs was again set to 200 with a learning rate of 0.0001, a weight decay of 0.001 and the Adam optimizer and a batch size of 32. For the selection of the negative and positive vector of the triplet loss, the temperature was set to 0.1. A loss based on the cosine similarity between anchors of the same diagnostic finding but with different states was added to the triplet loss to prevent the collapse of embeddings onto one state.

**Downstream Task Evaluation:** We evaluated the model performance on a test dataset with 260 pairs of 3D cardiac CT images and their respective radiology reports. The downstream tasks are the classification of calcification, stenosis, anomalies in the myocardium, and a dilated aorta or pulmonary trunk. Anomalies in the myocardium include a thickened myocardium, a hypertrophy in the myocardium or a myocarditis. Note that for our approach no additional classification network and training is necessary when a set of codebook vectors in the text feature space for every concept is available. We chose 40 codebook vectors for every state of every diagnostic finding. We compare our method with four state of the art alignment algorithms, CLIP (Radford et al., 2021a), ConVIRT (Zhang et al., 2022), GLoRIA (Huang et al., 2021) and another multi-instance alignment method (LSE+NL) (Wang et al., 2023). Note that GLoRIA, LSE+NL as well as MApLe work in a zero-shot setting on the downstream tasks, while ConVIRT and CLIP need fine-tuning and a small classification network for every tasks. For CLIP we utilized the ViT-L-14 implementation of the text and image encoder. Since this model is trained to work on 2D RGB images with a spatial size of $336 \times 336$, our 3D images with a volume of $512 \times 512 \times 512$ had to be adapted. All slices were taken in the axial direction and repeated three times. Afterwards every slice was rescaled to a spatial size of $336 \times 336$ and given to the CLIP image encoder.

The calculated embeddings were aggregated with a mean average pooling. For the Con-VIRT implementation, we utilized the BioClinical BERT model (Alsentzer et al., 2019) and the ResNet 3D implementation (He et al., 2016). Since we utilized the 3D implementation, the images had to be downsampled to a voxel size of $2 \times 2 \times 2$. For the GLoRIA (Huang et al., 2021) implementation, we also utilized the BioClinical BERT model. Since the code is also designed for 2D images, we processed our 3D volumes in the same way as for the CLIP model. For the multi-instance baseline method (Wang et al., 2023), we also utilized the BioClinical BERT model and splitted the 3D volumes into cubes with a size of 24 in every direction. Since we used 3D volumes, the images again had to be downsampled to a voxel size of 2.

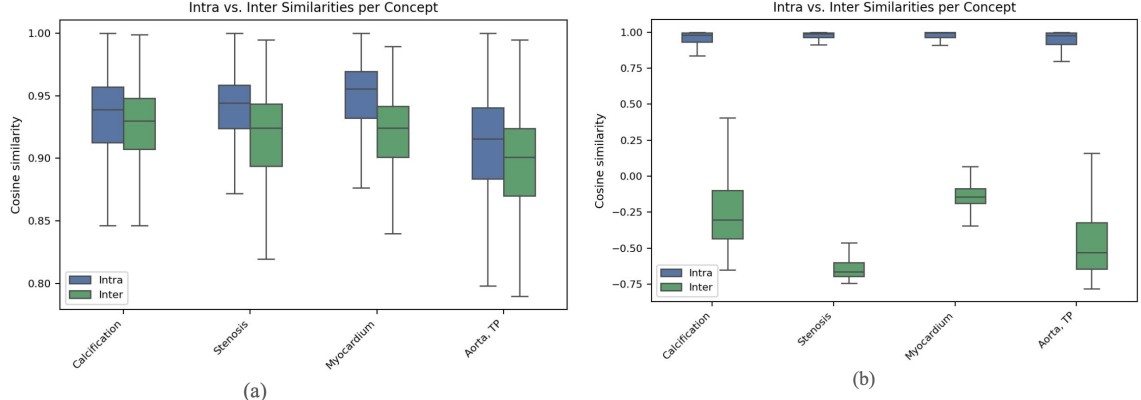

Figure 2: Boxplots of the mean pairwise cosine-similarities of sentences within each diagnostic finding between and across different states of this finding for (a) the BioClinical BERT model (Alsentzer et al., 2019) and (b) our fine-tuned text encoder.

## 5. Results

### 5.1. Report Embedding

We evaluated the capability of the BioClinical BERT model (Alsentzer et al., 2019) to detect small, diagnostically relevant aspects in sentences compared to our fine-tuned MApLE text encoder. For each diagnostic finding, we used the mean cosine-similarity between the same state of this finding and across the two states. Ideally, the cross-similarity should be low, i.e. close to -1, and the similarity between the same state close to 1. As shown in Figure 2, the cross-similarity for each finding is almost as high as the similarity between the same state, making alignment training with large 3D volumes difficult. In contrast, our fine-tuned text encoder has a feature space where the similarity across different states of findings is close to -1, whereas the similarity between the same finding is close to 1. This makes it easier for the alignment algorithm to identify clinically relevant differences in the sentences and thereby in the image.

## 5.2. Downstream Task Classification

We evaluated our feature extraction approach on four different downstream classification tasks and compared the results to SOTA alignment approaches and their classification results. The downstream tasks correspond to the important clinical findings in the report, i.e. calcification or stenosis in the coronary artery, anomalies in the myocardium like hypertrophy or a thickened myocardium, and a dilated aorta or pulmonary trunk (TP). The results can be seen in Table 1. We evaluated the results in terms of accuracy, F1-score, sensitivity, specificity, positive-predictive-value (PPV), negative-predictive-value (NPV) and area-under-the-curve (AUC). Note that since most of the methods do not work in a threshold-based setting, we could only compute a linear approximation of the AUC based on one sensitivity-specificity pair. For the calcification task, our model achieved the highest PPV with 56.95%. CLIP and LSE+NL have the highest sensitivity with 100%, ConVIRT the highest accuracy (58.33%), F1-score (70.00%) and NPV (77.78%) and GLoRIA the highest specificity with 52.86%. The highest AUC is achieved by ConVIRT at 57.30%, and almost identical by MApLe at 57.19%. For the stenosis task, our model achieved the highest F1-score with 37.71% as well as the highest PPV with 30% and the highest AUC with 55.64%. The highest accuracy of 79.17%, highest specificity of 100%, and highest NPV of 79.17% is achieved by GLoRIA, and the highest sensitivity of 100% by CLIP and LSE+NL. When classifying anomalies in the myocardium, CLIP and ConVIRT have the highest accuracy (90.28%) and specificity (100%), while GLoRIA achieves the highest F1-score (21.05%) and NPV (91.67%) as well as AUC (56.69%). The highest sensitivity of 100% is achieved by LSE+NL. For the classification of a dilated aorta or pulmonary trunk, our model again achieves the highest F1-score with 41.12%, the highest PPV with 33.85%, the highest NPV with 89.74% and the highest AUC with 66.33%. The highest sensitivity is present for GLoRIA with 100%. CLIP, ConVIRT and LSE+NL achieve the highest accuracy with 80.56% and the highest specificity with 100%.

## 6. Conclusion and Discussion

We propose a multi-instance alignment model to align embeddings from large-scale 3D images with multiple sentence embeddings, creating a multi-task model. Our model outperforms baseline models for stenosis classification and dilated aorta or pulmonary trunk classification, achieving F1-scores of 37.71% and 41.12%, respectively. Baseline models sometimes achieve higher sensitivity or specificity values by collapsing on one class, making them unsuitable for clinical use. While ConVIRT has the highest F1-score for calcification, its low specificity (20%) limits its effectiveness. Our model's F1-score of 60.78% on the other hand has a more balanced sensitivity and specificity and an almost identical AUC value as ConVIRT. Additionally, ConVIRT is, in contrast to our model, not zero-shot capable but had to be fine-tuned for each task separately, which likely resulted in the unbalanced sensitivity and specificity. MApLe was capable of outperforming the other two zero-shot approaches (GLoRIA and LSE+NL) on this task. All models fail to confidently classify anomalies in the myocardium. This is probably due to the heterogeneous nature of myocardial anomalies, which include the visually diverse conditions myocardial thickening, a hypertrophy in the myocardium or a myocarditis that manifest differently in imaging. A potential direction to improve performance would be to stratify myocardium abnormalities

| Method | Zero-shot | Acc. | F1 | Sens. | Spec. | PPV | NPV | AUC |
|---|---|---|---|---|---|---|---|---|
| **Calcification** | | | | | | | | |
| CLIP (Radford et al., 2021a) | ✗ | 0.5139 | 0.6789 | 1.0 | 0.0 | 0.5139 | 0.0 | 0.5 |
| ConVIRT (Zhang et al., 2022) | ✗ | 0.5833 | 0.7000 | 0.9459 | 0.2000 | 0.5556 | 0.7778 | 0.5730 |
| GLoRIA (Huang et al., 2021) | ✓ | 0.4722 | 0.4493 | 0.4189 | 0.5286 | 0.4844 | 0.4625 | 0.4738 |
| LSE+NL (Wang et al., 2023) | ✓ | 0.5139 | **0.6789** | 1.0 | 0.0 | 0.5139 | 0.0 | 0.5 |
| MApLe (ours) | ✓ | **0.5731** | 0.6078 | 0.6515 | 0.4922 | 0.5695 | 0.5780 | **0.5719** |
| **Stenosis** | | | | | | | | |
| CLIP (Radford et al., 2021a) | ✗ | 0.2083 | 0.3448 | 1.0 | 0.0 | 0.2083 | 0.0 | 0.5 |
| ConVIRT (Zhang et al., 2022) | ✗ | 0.7500 | 0.0526 | 0.0333 | 0.9386 | 0.1250 | 0.7868 | 0.4860 |
| GLoRIA (Huang et al., 2021) | ✓ | **0.7917** | 0.0 | 0.0 | 1.0 | 0.0 | 0.7917 | 0.5 |
| LSE+NL (Wang et al., 2023) | ✓ | 0.2083 | 0.3448 | 1.0 | 0.0 | 0.2083 | 0.0 | 0.5 |
| MApLe (ours) | ✓ | 0.5808 | **0.3771** | 0.5077 | 0.6051 | 0.3 | 0.7867 | **0.5564** |
| **Myocardium Anomalies** | | | | | | | | |
| CLIP (Radford et al., 2021a) | ✗ | 0.9028 | 0.0 | 0.0 | 1.0 | 0.0 | 0.9028 | 0.5 |
| ConVIRT (Zhang et al., 2022) | ✗ | 0.9028 | 0.0 | 0.0 | 1.0 | 0.0 | 0.9028 | 0.5 |
| GLoRIA (Huang et al., 2021) | ✓ | 0.7917 | **0.2105** | 0.2857 | 0.8462 | 0.1667 | 0.9167 | **0.5659** |
| LSE+NL (Wang et al., 2023) | ✓ | 0.0996 | 0.1811 | 1.0 | 0.0 | 0.0996 | 0.0 | 0.5 |
| MApLe (ours) | ✓ | **0.8731** | 0.0571 | 0.0384 | 0.9658 | 0.1111 | 0.9004 | 0.5021 |
| **Aorta, TP** | | | | | | | | |
| CLIP (Radford et al., 2021a) | ✗ | 0.8056 | 0.0 | 0.0 | 1.0 | 0.0 | 0.8056 | 0.5 |
| ConVIRT (Zhang et al., 2022) | ✗ | 0.8056 | 0.0 | 0.0 | 1.0 | 0.0 | 0.8056 | 0.5 |
| GLoRIA (Huang et al., 2021) | ✓ | 0.1944 | 0.3256 | 1.0 | 0.0 | 0.1944 | 0.0 | 0.5 |
| LSE+NL (Wang et al., 2023) | ✓ | **0.8056** | 0.0 | 0.0 | 1.0 | 0.0 | 0.8056 | 0.5 |
| MApLe (ours) | ✓ | 0.7577 | **0.4112** | 0.5238 | 0.8028 | 0.3385 | 0.8974 | **0.6633** |

Table 1: Classification results of our zero-shot approach on our internal 3D heart CT dataset compared to SOTA alignment methods CLIP (Radford et al., 2021a), ConVIRT (Zhang et al., 2022), GLoRIA (Huang et al., 2021) and LSE+NL (Wang et al., 2023). Zero-shot methods are models that don't have to be fine-tuned after the alignment, in contrast to the others that have to be fine-tuned for every single prediction task separately. The highest accuracy, F1-score and AUC among the zero-shot methods are marked.

into more homogeneous subcategories, for example by focusing on the most common abnormality types. Additionally, we only have a low number of 26 positive out of 260 samples in our test dataset. Future work should focus on improving this classification. The low PPV for the aorta/pulmonary trunk task despite the decent specificity (80%) likely stems from the imbalanced dataset, as a dilated aorta or pulmonary trunk is rare (42 positive samples out of 260 samples). The same applies to the PPV at 30% for stenosis, where we only have 65 positive samples out of 260. The sensitivity of 50.77% and specificity of 60.51% for the stenosis classification is probably caused by varying descriptions from doctors. For instance, some report a small, non-significant stenosis of 20% in the coronary arteries, while others omit it since it falls below the 50% threshold. Similarly, the same expression is often used for small or moderate calcification when not quantitatively measured.

In general, our method MApLe improved the classification, while several baselines collapse

to trivial solutions by predicting nearly all samples as positive or negative. In contrast, MApLe consistently avoids such collapse and yields balanced sensitivity–specificity trade-offs across the tasks. This is particularly evident for stenosis and aorta/tp, where MApLe is the only method achieving non-trivial performance. For calcification, the F1-score of one baseline is driven by a majority-class prediction, which limits the usability in clinical practice. Additionally, we have shown that even an encoder fine-tuned on medical texts struggles to differentiate between diagnostic findings in specialized reports like cardiac radiology. Our encoder stretched the text embeddings to ease alignment. Our image-patch encoder, conditioned on anatomical regions, improves the detection of small anomalies like stenosis or dilated aorta/pulmonary trunk in large 3D volumes. Overall, we have shown that our model MApLe is capable of aligning image representations with several sentences describing different diagnostic findings, even when the diagnostic findings are only small manifestations in a large 3D image. Further validation on a larger, multi-center dataset will be necessary to evaluate the generalization capability of our method.

## Acknowledgments

This research has been partially funded by the European Commission under Grant Agreement 101080302 AI-POD, by the Vienna Science and Technology Fund (WWTF, PREDICTOME) [10.47379/LS20065], from the European Union's Horizon Europe research and innovation programme under grant agreement No.101136299 — ARTEMIs, and in part by the Austrian Science Fund (FWF) under grants 10.55776/PIN3584324 and P35189 ONSET. This research was partially conducted within an Inter-University Cluster Project jointly funded by the University of Vienna and the Medical University of Vienna.

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
