# OpenReview forum: "MApLe: Multi-instance Alignment of Diagnostic Reports and Large Medical Images"
_MIDL.io/2026/Conference — MIDL 2026 Poster_

### Official Review · Reviewer_GzyV · 2025-12-29

**Confidence:** 4
**Preliminary Rating:** 3
**Final Rating:** 4

**Summary:**

This paper proposes MApLe, a multiple-instance multi-task vision-language alignement method tailored to cardiac CTs. The method finetunes finding-specific text encoders to amplify semantically meaningful differences in sentences, encodes image patches over anatomical segmentations, and align image patches and report sentences via a multiple-instance triplet loss. The authors demonstrated their method performs better on some but overall comparable performance to baselines on a set of 3D cardiac CT scans.

**Strengths:**

- The motivation is pretty clear, i.e., use local encoding since some radiology findings appear to be tiny on radiograph and is very imoprtant.
- I also like there is finding specific coding of sentences in the report. It kind of make sense to just say since we are working with a limited set of findings then let's just make sure the text embedding work well on this subset of English language.

**Weaknesses:**

+ This paper argues their approaches improve "alignemnt performance" compared to SoTA baseline models. However, the authors evaluated on multi-task classification only and did not really alignment performance, e.g., how well can we locate referred expression in images or ground findings in radiographs. The authors should either refrain from making such a claim (and focus on classification) or measure alignment performance (e.g,, refer to BioViL's proposed overlap metric) and compare with baselines.
+ Although MApLe improves F1-score for some tasks, absolute performance remains modest, and baselines sometimes outperform MApLe. The baseline is quite old as well (circa. 2021). Maybe a more comprehensive evaluation is required to provide a more reliable measure of performance.

**Detailed Comments:**

+ Writing could be improved. There are too many variables introduced in Section 3 with subscripts and superscripts, it would help reader with understanding the method if you could just define variables when it's absolutely needed / important - pick a few and really emphasize what they meant. More rigor in how the variables are defined would help too, e.g., if $f_{an}$ condition on anatomical region $a\in A$, why is the function $p\mapsto x$ instead of $(p,a)\mapsto x$ and why is it $f_{an}$ instead of $f_a$? It would also clarify things if the authors could help the reader distinguish between scalar, vector, and functions.
+ Related to previous point, might be good to add important variables introduced in Section 3 to Figure 1 so people have an easier time parsing the method section.
+ There are prior work that relies on multiple instance learning for vision language alignment, e.g., [1,2]. Be good to discuss these related works and how this paper is similar/different to these prior works.



[1] 2023, Using Multiple Instance Learning to Build Multimodal Representations

[2] 2024, Pathm3: A multimodal multi-task multiple instance learning framework for whole slide image classification and captioning

**Justification Of Final Rating:**

The authors clarified that "alignment" is more about representation alignment than otherwise. The authors also clarified their contribution and supplied additional baselines for comparison. Therefore I'll increase rating to weak accept.

**Justification Of The Preliminary Rating:**

The paper addresses a relevant and clinically motivated problem and proposes a reasonable multiple-instance vision–language alignment framework. The finding-specific text encoding is intuitive and technically sound. However, the main claim of improved alignment is not directly evaluated, as experiments focus only on downstream classification without localization or grounding metrics. Performance gains are mixed and often comparable to older baselines. Clearer validation of alignment quality and stronger positioning relative to prior MIL-based multimodal work are needed, making this a borderline submission.

**Questions To Address In The Rebuttal:**

See weaknesses and comments.

---

> ### Author Response · Authors · 2026-01-24
>
> We thank the Reviewer for the thorough evaluation of our paper and pointing out issues that need clarification. We have considered these points and would like to offer our response as follows (changes based on your comments are highlighted in the submitted revision):
>
> 1. Interpreting the results and modest overall performance (also mentioned by Reviewer 1):
>
> We clarify in the revised manuscript that only MApLe, GLoRIA and LSE+NL are zero-shot approaches, while ConVIRT and CLIP are not. We have high-lighted this in the table to facilitate taking this into account, as the non-zero-shot approaches are not directly comparable, but informative regarding the strengths of the method.
>
> The variability of the results is due to the radiological findings differing in scale, frequency, and imaging characteristics. Overall zero-shot learning of the alignment between findings in reports and images is a hard problem, since findings are often relatively small, or subtle compared to the overall cross-sectional variability of anatomical structures. Therefore, we believe that the present results demonstrate feasibility, and the benefits of the proposed approach. In addition they allow for assessing their level with other fine-tuning based methods.
>
> While classification performance varies across tasks, MApLe consistently achieves the most balanced performance in the evaluation metrics across the tasks. Each baseline approach collapses (classifying to a single class) for multiple tasks, and the myocardium task remains challenging for all methods (see below). Collapse is particularly evident for stenosis and aorta, where MApLe is the only method achieving non-trivial performance. For calcification, ConVIRT performs comparably well (F1), but unbalanced (sens., spec.) suggesting a role of the fine-tuning in this result.
>
> 2. Comparison with baseline approaches:
>
> We have added an additional baseline approach in the comparative evaluation (Wang et al. 2023). While some baseline models were originally proposed around 2021, they remain widely used and competitive references for vision–language learning in medical imaging. In particular, CLIP-style and contrastive pretraining approaches form the conceptual foundation of many more recent methods, and are still commonly adopted. We therefore selected these baselines as strong, well-established points of comparison for our setting. We are adding an explanation in the manuscript.
>
> 3. Clarification of the use of “alignment performance”:
>
> We thank the reviewer for pointing out this ambiguity. In this work, we use the term “alignment” to refer to representation-level alignment between image and text feature spaces, as commonly adopted in CLIP-style vision–language models, rather than explicit spatial grounding or localization. This representation alignment performance can be induced from the downstream classification, since the classification tasks correspond to the conditions stated in the report with which the image features should be aligned. We agree that our evaluation focuses on downstream multi-task classification and does not measure localization-based alignment metrics (e.g., overlap or grounding scores), which would require region-level annotations that are not available in our setting. To avoid confusion and over-claiming, we will revise the manuscript to clarify this distinction and rephrase our claims to focus on representation-level alignment as a pre-training method and its effect on downstream performance.
>
> 4. Figure 1: We appreciate the Reviewer’s suggestion to include variable names in our visualization figure. The revised submission contains this improvement of the figure.
>
> 5. Prior multi-instance learning: We thank the reviewer for pointing out these relevant works. We agree that multiple instance learning (MIL) has been explored in multimodal settings. The work in [1] addresses image–text alignment but is limited to 2D natural images and does not consider 3D clinical imaging. The framework in [2] employs MIL in a multimodal context for whole-slide image classification and caption generation; however, it does not aim at aligning image and text representations in a shared embedding space, but rather focuses on generative captioning objectives. We will include and discuss both works in the related work section. Additionally, we added [1] as an additional baseline.
>
> 6. Section 3: We thank the reviewer for the writing suggestions in section 3 to make it more clearer. We incorporated these suggestions in the submitted revision.

---

### Official Review · Reviewer_pDnM · 2026-01-10

**Confidence:** 4
**Preliminary Rating:** 2
**Final Rating:** 4

**Summary:**

The authors propose MApLe, a multi-instance vision-text alignment framework that locally aligns image patches with their corresponding diagnostic sentences and states within the radiology report. The goal is to align smaller, often missed features in reports to image patches. Their approach utilizes triplet loss, using an attention-pooled image embedding as anchor and corresponding positive and negative sentence embeddings of the same diagnostic state (only the former matches the diagnostic state). They conduct extensive experiments on an institutional dataset and compare with baselines. Results suggest higher F1 scores on a subset of tasks.

**Strengths:**

- The authors tackle a very important aspect of scaling alignment methods for 3D volumes, where the trade-off between computational efficiency and performance is a critical consideration.
- The experiments are extensive and well-designed.
- The methods are well-described and sufficient detail is provided for reproducibility.

**Weaknesses:**

- I recommend the authors reorganize section 4.2 to make it easier to read, as it includes no paragraph nor section breaks.
- Expanding upon the prior point, the results could also be reworked to reduce receptiveness.
- What are the characteristics of the private dataset? How were the images acquired, etc.? If there is relevant publication on this dataset, the authors should cite it. Otherwise, I suggest including these details in appendix.
- The results seem to not demonstrate any reliable trend, making it hard to justify whether the proposed method is an improvement over baselines.

**Detailed Comments:**

- Describing radiology reports as "semi-structured" and "consistent" might be an overstatement, given that report structure can vary dramatically between scanning protocols (see, standardized mammogram reports), radiologists, and institutions.
- What parts of the report were considered? Was the report treated as a single free-text entry or specific sections (e.g., findings) extracted?
- Visualization of image embedding in Fig. 1 could be improved for further clarity as it is not immediately clear the process is identical to sentence embedding (i.e., relying on a classifier to predict region)
- Paper has some minor typos (e.g., "During text-time, [...]" in section 3.3)
- Do the authors intend to release the code?

**Justification Of Final Rating:**

I thank the authors for providing much needed clarity to the paper and presented results. My concerns have mostly been addressed, particularly related to clarity and interpretability of results (especially the trivial collapse discussion). Therefore, I recommend acceptance.

**Justification Of The Preliminary Rating:**

While the method is quite interesting and well-motivated, it is difficult to recommend acceptance for this given the poor paper organization, missing details on training data, and inconsistent results that don't suggest that the proposed method outperforms baselines.

**Questions To Address In The Rebuttal:**

Please see weakness and detailed comments.

---

> ### Author Response · Authors · 2026-01-24
>
> We appreciate the Reviewer’s feedback, and are grateful for pointing out important issues that need clarification. We have considered these points and would like to offer our response as follows. Changes based on the comments are highlighted in the submitted revision:
>
> 1. Interpreting the heterogeneity of results (also mentioned by Reviewer 1):
>
> We clarify in the revised manuscript that only MApLe, GLoRIA and LSE+NL are zero-shot approaches, while ConVIRT and CLIP are not. We have high-lighted this in the table to facilitate taking this into account, as the non-zero-shot approaches are not directly comparable, but informative regarding the strengths of the method..
>
> The variability of the results is due to the radiological findings differing in scale, frequency, and imaging characteristics. While classification performance varies across tasks, MApLe consistently achieves the most balanced performance in the evaluation metrics across the tasks. Each baseline approach collapses (classifying to a single class) for multiple tasks, and the myocardium task remains challenging for all methods (see below). Collapse is particularly evident for stenosis and aorta, where MApLe is the only method achieving non-trivial performance. For calcification, ConVIRT performs comparably well (F1), but unbalanced (sens., spec.) suggesting a role of the fine-tuning in this result.
>
> 2. Dataset description:
>
> We thank the reviewer for pointing out missing information in the paper. The data stems from an internal retrospective patient cohort from the Vienna General Hospital between 2016 and 2023. In the submitted revision, a more detailed description about the characteristics of our utilized dataset is given.
>
> 3. Clarification “semi-structured”/consistent:
>
> We agree that radiology reports are not formally structured and that report organization can vary substantially across radiologists, protocols, and institutions. Our intention was not to suggest a fixed or standardized report structure. Instead, we refer to a form of semantic consistency: for a given imaging modality and clinical task (e.g., cardiac CT), radiologists are typically required to describe the same anatomical structures and relevant abnormalities. As a result, while reports may differ in ordering and phrasing, sentences referring to the same anatomical regions or findings often exhibit similar semantic content and similar expressions. We will revise the wording to avoid the terms “semi-structured” and “consistent” and clarify this distinction in the revised manuscript.
>
> 4. Clarification of the report processing: The reports are provided as unstructured free text without explicit sections. We therefore apply sentence-level filtering based on task-specific keywords to retain clinically relevant descriptions while removing acquisition-related or administrative sentences. This removes non-clinical sentences, ensuring that the model still needs to learn the association between visual regions and textual descriptions. We will clarify this in the revised manuscript.
>
> 5. Fig.1: We thank the reviewer for pointing out missing information in the visualization of our method. In the revised rebuttal document, the figure now includes the patch extraction method.
>
> 6. Code release: Yes, we will release the code.
>
> 7. Structure and typos: We thank the reviewer for pointing out structural issues as well as typos in the paper. We revised our submission to make the structure clearer and to eliminate the typos.

---

### Official Review · Reviewer_ebjV · 2026-01-10

**Confidence:** 3
**Preliminary Rating:** 3
**Final Rating:** 4

**Summary:**

This paper introduces MApLe, a multi-instance vision-language alignment method that links specific sentences in radiology reports to localized anatomical regions in 3D medical images. The approach combines a finely-tuned text encoder for capturing fine-grained diagnostic differences, anatomical region-conditioned patch-based image encoders, and a multi-task alignment objective using triplet loss.  Experiments on a 3D cardiac CT dataset show that MApLe is competitive in zero-shot classification of findings like stenosis and aortic dilation against methods including CLIP, ConVIRT, and GLoRIA.

**Strengths:**

1. The paper is organized and clearly explains the idea.
2. The approach is well-structured,  combining a fine-tuned BERT for text, a patch-based 3D image encoder conditioned on anatomy, and a task-specific alignment mechanism.
3. The paper provides a comprehensive comparison of MApLe against strong baselines (CLIP, ConVIRT, GLoRIA) on multiple clinical tasks and reports a full suite of metrics.

**Weaknesses:**

1. Although MApLe shows improvements in some tasks like stenosis and aorta, the overall performance remains modest compared to other methods. For tasks like calcification and myocardium, it does not consistently outperform all baselines.
2. The model is trained on an internal and relatively small dataset of 768 training samples. It would be good to discuss the generalizability and scalability of MApLe.

**Detailed Comments:**

1. Table 1 is referenced as "Table 5.2" in section 5.2. Please correct the reference.
2. Duplicate sentence “To tackle these challenges, …” in Section 1. Please remove the duplicate.
3. In section 3.3, it would be good to further clarify the concept of “a positive embedding that is dissimilar”.

**Justification Of Final Rating:**

I thank the authors for their response. This mostly addresses my concerns regarding paper clarity, interpretation of the results, and the limitation of dataset size. Given these improvements, I have changed my rating to weak accept.

**Justification Of The Preliminary Rating:**

The paper presents a novel approach for fine-grained vision-language alignment in 3D medical imaging, addressing an important gap in linking textual findings to localized anatomical regions. Its strengths lie in its clear formulation, structured approach, and thorough evaluation against the baselines. However, the empirical gains are modest, and performance is inconsistent across tasks, raising questions about its current clinical utility. Additionally, the reliance on a small, domain-specific dataset limits confidence in generalizability.

**Questions To Address In The Rebuttal:**

1. Could the authors comment on the effects of dataset size and domain specificity on generalizability? Are there plans to validate on larger datasets?
2. The model performs poorly on myocardium anomaly classification. Can the authors hypothesize why this task is particularly challenging and how the method might improve performance?

---

> ### Author Response · Authors · 2026-01-24
>
> We appreciate the Reviewer's thorough assessment and detailed comments on our paper. In the following we address the concerns point by point.  We have highlighted changes based on the comments in the resubmitted revision of the manuscript.
>
>
> 1. Interpreting the heterogeneity of results:
>
> MApLe, GLoRIA and LSE+NL are zero-shot approaches, while ConVIRT, and CLIP are not. We have high-lighted this in the table to facilitate taking this into account, as the non-zero-shot approaches are not directly comparable, but informative regarding the strengths of the method..
> The radiological findings differ in scale, frequency, and imaging characteristics. While classification performance varies across tasks, MApLe consistently achieves the most balanced performance in the evaluation metrics across the tasks. Each baseline approach collapses (classifying to a single class) for multiple tasks, and the myocardium task remains challenging for all methods (see below). Collapse is particularly evident for stenosis and aorta, where MApLe is the only method achieving non-trivial performance. For calcification, ConVIRT performs comparably well (F1), but unbalanced (sens., spec.) suggesting a role of the fine-tuning in this result.
>
> 2. Myocardium performance:
>
> All methods perform poorly on the task to classify if anomalies in the myocardium are present. We hypothesize that this is due to the heterogeneous nature of myocardial anomalies, which include the visually diverse conditions myocardial thickening, a hypertrophy in the myocardium or a myocarditis that manifest differently in imaging. A potential direction to improve performance would be to stratify myocardium abnormalities into more homogeneous subcategories, for example by focusing on the most common abnormality types.
>
>
> 3. Dataset size and corresponding limitations:
>
> We acknowledge that MApLe is trained on a single center dataset with 768 samples, which limits direct claims about generalizability. We have clarified this in the revised manuscript. To the best of our knowledge, there are currently no public datasets pairing 3D medical imaging with radiology reports. MApLe is specifically designed for the realistic clinical setting including large 3D images with small or subtle diagnostically relevant findings. While domain specificity may affect transferability, the proposed region-conditioned alignment mechanism is not dataset-specific and should, in principle, scale to larger cohorts. We will clarify these limitations in the revised manuscript and on the longer term plan for further validation on larger, multi-center dataset that we are collecting at the moment.
>
>
> 4. Table reference and duplicate sentence: We thank the reviewer for pointing out these mistakes. They are revised in the submitted rebuttal document.
>
>
> 5. Dissimilar positive embedding: We acknowledge the need to further clarify the concept of dissimilar positive embeddings during the alignment training. Further clarification was added to the revised submission of the paper.

---

### Author Rebuttal · Authors · 2026-01-24

**Rebuttal:**

We thank the Reviewers for their constructive feedback. While there is consensus that the multi-instance vision-language alignment approach is novel and addresses a relevant gap, the reviewers pointed out the need for clarification in several aspects including results and data. We summarize the main points here, and provide details in the responses to the individual reviewers. Additionally, the revised submission can be found below.

1. Clarification of results heterogeneity:

The characteristics of the 4 radiological findings are very different in frequency, scale and appearance. Except for calcification with 132 positive out of 260 samples, distributions are unbalanced (stenosis: 25% positive, myocardium: 10%, aorta: 16%). While each baseline method collapses (sens., spec.) for at least 2 findings, MApLe yields more consistent and balanced results across findings. It outperforms all baselines for aorta and stenosis. This reflects the heterogeneous nature of radiological findings, and we will expand the discussion accordingly.

Not all baselines are zero-shot approaches. While MApLe, GLoRIA and LSE+NL perform zero-shot learning, CLIP and ConVIRT do not.

Following the suggestion of a reviewer, we have added another more recent baseline method to the comparison experiments (Table 1).

2. Data set details and method generalization:

Although the data set is of moderate size (n=768), we believe results are informative, as they reflect the heterogeneity of findings, and exhibit consistency. We are currently in the process of expanding the data set to multiple centers, but believe the current results carry value for the community. We will expand the data description, and clarify the corresponding limitation in the paper.

The code will be released.

**Supporting Material:**

/attachment/40ffd0070db67a50a7dc809f6614dad1e8e2c4b3.pdf

---

### Comment · Area_Chair_tNXX · 2026-01-30
**Reviewers, please take a moment to review the authors' rebuttal and revise your scores**

Update your final rating by clicking “Edit” → “Official Review” and providing the Final Rating by February 1st 2026 (23:59 AoE).

---

### Meta-Review · Area_Chair_tNXX · 2026-02-09

**Recommendation:** Accept (Poster)
**Confidence:** 5

**Metareview:**

This manuscript describes a multi-task, multi-instance vision-language alignment approach that improves alignment between fine-grained sentence-level text embeddings and image patches using a triplet loss. The reviewers noted that the work's motivation is clear and that the authors have provided comprehensive experiments. The revised manuscript largely addresses the requested clarifications from the reviewers, though the presented results and improvements remain modest.

---

### Decision · Program_Chairs · 2026-02-13

Accept (Poster)